# Emergent hypernetworks in weakly coupled oscillators

Eddie Nijholt[1], Jorge Luis Ocampo-Espindola[2], Deniz Eroglu[3], István Z. Kiss[2] &
Tiago Pereira ®[1,4] ✉

Networks of weakly coupled oscillators had a profound impact on our understanding of complex systems. Studies on model reconstruction from data have shown prevalent contributions from hypernetworks with triplet and higher interactions among oscillators, in spite that such models were originally defined as oscillator networks with pairwise interactions. Here, we show that hypernetworks can spontaneously emerge even in the presence of pairwise albeit nonlinear coupling given certain triplet frequency resonance conditions. The results are demonstrated in experiments with electrochemical oscillators and in simulations with integrate-and-fire neurons. By developing a comprehensive theory, we uncover the mechanism for emergent hypernetworks by identifying appearing and forbidden frequency resonant conditions. Furthermore, it is shown that microscopic linear (difference) coupling among units results in coupled mean fields, which have sufficient nonlinearity to facilitate hypernetworks. Our findings shed light on the apparent abundance of hypernetworks and provide a constructive way to predict and engineer their emergence.

Networks of weakly coupled oscillators are prolific models for a variety of natural systems ranging from biology[1,2] and chemistry[3,4] to neuroscience[5,6] via ecology[7] to engineering[8]. Such networks serve as stepping stones to understand collective dynamics[9–12] and other emergent phenomena in networks[13,14]. In these models, the interactions are described in a pairwise manner and the collective dynamics of a network can be predicted by the superposition of such pairwise interactions.

Recent work, however, suggests that many networks described as pairwise interactions can be better described in terms of hypernetworks with triplet and quadruplet interactions among nodes[15–18]. In fact, hypernetworks appear as suitable representations of certain dynamical processes found in physics[19,20], chemistry[21] and neuroscience[22,23]. This has ignited research aimed at understanding the impact of higher-order interactions on the dynamical behavior of complex systems[24–27]. Moreover, besides considering hypernetworks as a good description of such models, we observed that hypernetworks could be revealed in data-driven model reconstructions when the original model is a network. Therefore, a major puzzle is why hypernetworks emerge as the fitting description of actual network data.

Here, we show that hypernetworks can describe experimental data of networks of electrochemical oscillators with nonlinear coupling. We uncover a mechanism that generates higher-order interactions as a model to describe oscillator networks from data. First, we show that sparse model recovery from data reveals higher-order interactions. We then develop a theory for the emergence of such higher-order interactions when the isolated system is close to a Hopf bifurcation. We provide an algorithm to reveal emergent hypernetwork and its emergent coupling functions for any network in disciplines ranging from neuroscience to chemistry. The emergent hypernetworks provide a dimension reduction that allows the characterization of critical transitions.

[1]Instituto de Ciências Matemáticas e Computação, Universidade de São Paulo, São Carlos, Brazil. [2]Department of Chemistry, Saint Louis University, St. Louis, MO, USA. [3]Faculty of Engineering and Natural Sciences, Kadir Has University, Istanbul, Turkey. [4]Department of Mathematics, Imperial College London, London, UK. ✉e-mail: tiago.pereira@imperial.ac.uk

## Results

### Emergent hypernetworks in electrochemical experiments

We designed an experimental system with four oscillatory chemical reactions coupled with nonlinear feedback and delay arranged in a ring network (see Fig. 1a). The set-up consists of a multichannel potentiostat interfaced with a real-time controller and connected to a Pt counter, a $Hg/Hg_2SO_4$ sat $K_2SO_4$ reference, and four Ni working electrodes in 3.0 M sulfuric acid electrolyte. At a constant circuit potential ($V_0 = 1100$ mV with respect to the reference electrode) and with an external resistance ($R_{ind} = 1.0$ kohm) attached to each nickel wire, the electrochemical dissolution of nickel exhibits periodic current and electrode potential oscillations with a natural frequency of 0.385 Hz.

Without coupling, we adjusted the natural frequency of each oscillator to have a ratio with respect to oscillator 1 as $\omega_2/\omega_1 = 2.53$ ($\approx 2.5$), $\omega_3/\omega_1 = 1.56$ ($\approx 1.5$) and $\omega_4/\omega_1 = 2.53$ ($\approx 2.5$) with a set of resistors and capacitors ($C_{ind}$), see Supplementary Note 1.) The natural frequencies create opportunities for triplet resonances, as there are small detunings for $\omega_1 - \omega_2 + \omega_3$ and $\omega_1 - \omega_4 + \omega_3$, as well as pairwise resonances $\omega_2 \approx \omega_4$.

The individual electrode potentials ($E_k$) were recorded and rescaled and offset corrected

$$\tilde{E}_k = O_k[E_k - o_k], \tag{1}$$

where $o_k$ and $O_k$ are the time-averaged electrode potential and amplitude rescaling factor, respectively. (The rescaling factors, $O_k = 0.5, 1, 0.5, 1$ were applied to counter the different amplitudes of the slow oscillators.) A ring-coupling can be introduced with external feedback (see Fig. 1b, c) according to

$$V_k(t) = V_{0,k} + K \sum_{\ell=1}^{4} A_{k\ell} h[\tilde{E}_k(t), \tilde{E}_\ell(t - \tau)], \tag{2}$$

where $V_k(t)$ and $V_{0,k}$ are the applied and the offset circuit potential of the $k$th electrode, respectively, $K$ is the coupling strength, $A_{k\ell}$ is the adjacency matrix, $\tau$ is a time delay, and

$$h[\tilde{E}_k(t), \tilde{E}_\ell(t - \tau)] = (\tilde{E}_k(t) + \tilde{E}_k(t)^2)\tilde{E}_\ell(t - \tau). \tag{3}$$

This delayed nonlinear feedback modulates the impact of the coupled units with a bias towards positive values (similar to a diode operation in the (−1, 1) interval). Note that this form of feedback is fundamentally different from previously applied nonlinear schemes[4] in that it does not produce obvious synchronization patterns, for example, one and multi-cluster states.

Figure 1d shows the time series of the electrode potential for $K = 5.2$ and $\tau = 1.65$ s. The slow oscillators (1 and 3) have larger amplitudes and the time series exhibit nonlinear waveform modulations without any obvious synchronization pattern (one-cluster state).

From the potentials $\tilde{E}_k$ we extract the frequencies $\dot{\theta}_k$ and apply a first-order Savitzky-Golay filter with a time window of 45 s to remove the in-cycle and short-range phase fluctuation, as shown in Fig. 1e (solid line). For each oscillator, a slow variation is seen as the oscillators slow down and speed up on a timescale of about 100 s (or 40 cycles); notably, the elements 1 and 3 exhibit similar $\dot{\theta}_k$ oscillations, which are different from those in elements 2 and 4.

To describe the nature of the phase dynamics, we consider the slow triplet phase differences

$$\begin{aligned}
\phi_1 &= \theta_1 - \theta_2 + \theta_3 \\
\phi_2 &= \theta_1 - \theta_4 + \theta_3,
\end{aligned} \tag{4}$$

which correspond to the triplet frequency detunings.

The impact of triplet interactions on the dynamics can be extracted with a LASSO fit to

$$\dot{\theta}_k = \hat{\omega}_k(t) + \sum_{j=1}^{2} C_j^k \sin(\phi_j) + D_j^k \cos(\phi_j) \tag{5}$$

where $\hat{\omega}_k(t) = \hat{\omega}_k^0 + \hat{\omega}_k^1 t + \hat{\omega}_k^2 t^2$ is the fitted, slowly drifting (up to quadratic variation in time) natural frequency, and $C_j^k$ and $D_j^k$ are the amplitudes of the sin and cos phase coupling functions corresponding to the appropriate triplet phase differences. The strength of the triplet interactions $j = 1, 2$ (for $\phi_j$) on oscillator $k$ is given by the amplitudes $H_j^k = \sqrt{(C_j^k)^2 + (D_j^k)^2}$.

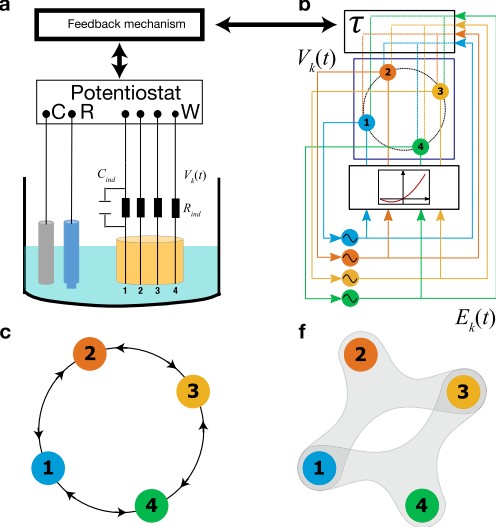
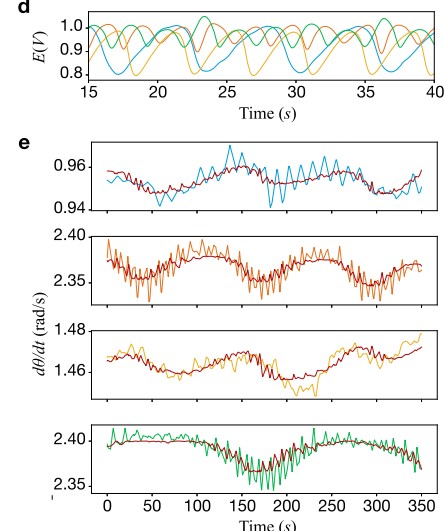

**Fig. 1 | Emergent hypernetworks in an electrochemical network experiment.**
**a** Experimental setup. **b** Schematic illustration of the electrochemical experiment with the nonlinear feedback. The blue, orange, yellow, and green lines represent the elements 1 to 4, respectively. The electrode potential signals ($E_k$) of the four (nearly) isolated electrodes are nonlinearly modulated and fed back with a delay $\tau$ to the corresponding circuit potential ($V_k$), which drives the metal dissolution. (The delay is implemented by storing the past data in the memory of the computer.) **c** Representation of the in a ring network topology used in the experiment. **d** Electrode potential time series. **e** Filtered and fitted (dark red line) instantaneous frequency using LASSO for hypernetwork reconstruction corresponding from top to bottom to oscillators 1 to 4, respectively. **f** Experimental recovery of the phase interactions given by a hypernetwork.

The dynamics of oscillators 1 and 3 are impacted by both triplet interactions; $\phi_1$ impacts oscillators 1 and 3 with amplitudes $4.9 \times 10^{-3}$ and $4.4 \times 10^{-3}$, and $\phi_2$ with $2.3 \times 10^{-3}$ and $3.2 \times 10^{-3}$, respectively. However, the dynamics of oscillators 2 and 4 are only impacted by triplet interactions $\phi_1$ (with amplitude $1.33 \times 10^{-2}$) and $\phi_2$ ($1.7 \times 10^{-2}$), respectively. These triplet interactions describe phase fluctuations over the long time scale (red curves in Fig. 1e). Therefore, we can conclude that the phase dynamics of the oscillators coupled in a ring can be described by a hypernetwork shown in Fig. 1f.

The fact that model recovery provides triplets as the best description is rather puzzling. Also given that the resonant behavior $\omega_2 \approx \omega_4$ did not appear in the model recovery from data. This suggests an interplay between the resonant frequencies and the network topology. The question arises, which resonances/triplet interactions emerge from a large number of possibilities in a given network, natural frequencies, and nonlinear coupling? An outstanding question is what is the origin of these triplet interactions that were generated by pairwise physical coupling?

## A theory for emergent higher-order interactions

To answer these questions, we develop a theory that captures the important characteristics of the experiments: nonlinear coupling and triplet resonance conditions. We consider the networks

$$\dot{z}_k = f_k(z_k) + \alpha \sum_{\ell=1}^{n} A_{k\ell} h_k(z_k, z_\ell) \tag{6}$$

where $z_k \in \mathbb{C}$ is the state of the $k$th oscillator, $h_k : \mathbb{C} \times \mathbb{C} \to \mathbb{C}$ is the pairwise coupling function, $A_{k\ell}$ is the adjacency matrix, and $\alpha > 0$ is the coupling strength. When the isolated system is close to a Hopf bifurcation, the dynamics is described by $f_k(z_k) = \gamma_k z_k + \beta_k z_k |z_k|^2$[228]. The Hopf bifurcation is a common route to oscillations in nonlinear systems and describes the appearance of oscillations in applications[2,3,5,6,8]. Our proofs are valid for $\gamma_k = \lambda + i\omega_k$ with small $\lambda$ and $\omega_k$ satisfying resonance conditions. We fix $\beta_k = -1$, but this value is immaterial. We develop a normal form theory to eliminate unnecessary terms of $h(z_k, z_\ell)$ and to expose higher-order ones that predict the dynamics. To a network of the form of Eq. (6) we associate non-resonance conditions that allow us to get rid of the leading interaction terms in $\alpha$.

Since $h(z_k, z_\ell)$ is a linear combination of monomials and the theory can be applied to each monomial independently, we assume first that $h(z_k, z_\ell)$ is a single monomial of the form

$$h(z_k, z_\ell) = z_k^{d_1} \bar{z}_k^{d_2} z_\ell^{d_3} \bar{z}_\ell^{d_4} \tag{7}$$

for non-negative numbers $d_1, \ldots, d_4$. Our major theoretical result is a formulation of a non-resonance condition given by

$$(d_1 - d_2 - 1)\omega_k + (d_3 - d_4)\omega_\ell \neq 0. \tag{8}$$

This condition shows up naturally in our approach, as a monomial Eq. (7) can only be eliminated by a transformation that divides by the left-hand side of Eq. (8). Hence, an interaction term in the coupling function $h$ given by Eq. (7) can only be removed if the non-resonance condition is satisfied. The non-resonance condition is defined as the union over all non-resonance conditions of its monomial terms. The network non-resonance conditions are given by the union over all non-resonance conditions of $h(z_k, z_\ell)$ for which $A_{k\ell} \neq 0$. Our result is the following:

In Methods, we show that given Eq. (6) with $h : \mathbb{C} \times \mathbb{C} \to \mathbb{C}$ a smooth map with vanishing constant terms, under the network non-resonance conditions, there is a coordinate transformation that

eliminates pairwise interaction terms and reveals the higher-order interactions. The proof consists of two main steps:

(i) Existence of a polynomial change of variables. Consider

$$u_k = z_k - \alpha P_k \tag{9}$$

for some polynomials $P_k$. The goal is to design $P_k$ such that in the variables $u_k$ interaction terms linear in $\alpha$ vanish. We obtain higher-order interactions of order $\alpha^2$. For Eq. (6) we use

$$P_k(z) = \sum_{\ell=1}^{n} A_{k\ell} \tilde{h}_{k\ell}(z_k, z_\ell), \tag{10}$$

where $\tilde{h}_{k\ell}(z, w)$ is the function obtained from $h(z, w)$ by transforming each monomial according to the following replacement rule:

$$z^{d_1} \bar{z}^{d_2} w^{d_3} \bar{w}^{d_4} \mapsto \frac{z^{d_1} \bar{z}^{d_2} w^{d_3} \bar{w}^{d_4}}{(d_1 - 1)\gamma_k + d_2 \bar{\gamma}_k + d_3 \gamma_\ell + d_4 \bar{\gamma}_\ell} \tag{11}$$

Note that the imaginary part of the denominator in Eq. (11) is precisely the left-hand side of Eq. (8). While bringing the equations to the new form, we face a major challenge to understand the combinatorial behavior of the Taylor coefficients during the transformation. We define a bracket on the space of polynomials to track these coefficients.

(ii) Dealing with transformed isolated dynamics. The second major challenge lies in the fact that another coordinate transformation is needed to eliminate terms coming from the isolated dynamics $f_k$. Indeed, as we eliminate coupling terms linear in $\alpha$, other terms linear in $\alpha$ appear due to the isolated dynamics. A remarkable fact is that the same non-resonance conditions also ensure that the second transformation exists.

Our theorem is applicable to a much broader class of coupling functions and network formalisms than what is described by Eq. (6). A rich variety of new interaction rules can emerge, depending on the specifics of the set-up (see Supplementary Note 2).

Applying the replacement rule Eq. (11) we obtain

$$\dot{u}_k = f_k(u_k) - \alpha^2 \left\{ \sum_{\ell=1}^{n} \sum_{p=1}^{n} \left[ A_{k\ell} A_{kp} \, {}^1G_k^{\ell p}(u_k, u_\ell, u_p) - A_{k\ell} A_{\ell p} \, {}^2G_k^{\ell p}(u_k, u_\ell, u_p) \right] \right\}, \tag{12}$$

up to higher-order terms in $\alpha$ and $u$. In Methods, we discuss the new coupling functions ${}^1G_k$ and ${}^2G_k$ some their properties. The coupling is now $\alpha^2$ explaining anomalous synchronization transitions that appears in networks (see Supplementary Note 3).

## Emergent hypernetworks explain experimental data

Similar to the experiments we consider a ring of four oscillators with coupling function

$$h(z, w) = z\bar{w} + z^2 \bar{w}. \tag{13}$$

Instead of delay, the oscillators are coupled through a conjugate variable that enables a streamlined theoretical treatment. Close to a Hopf bifurcation, the delay would have an effect of advancing the oscillations over half a period. As before, we consider $\omega_1 - \omega_2 + \omega_3$ and $\omega_1 - \omega_4 + \omega_3$ to be close to zero, so, capturing the triplet resonance in the experiments. We apply our theory to this case to unravel how higher-order interactions appear in the data.

The coupling function is a combination of $z\bar{w}$ and $z^2\bar{w}$, providing $d_1 = 1$ and $d_4 = 1$ for the first monomial and $d_1 = 2$ and $d_4 = 1$ for the latter.

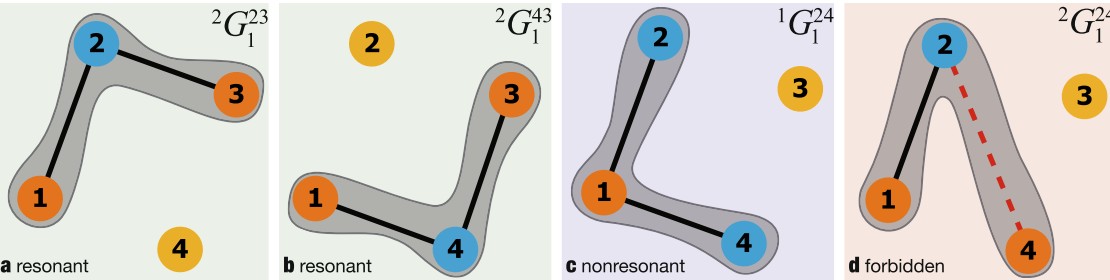

**Fig. 2 | Emergent higher-order interactions from the original ring network.**
Coupling functions appearing in Eq. (12) of node 1. Colors correspond to signs in
the phase combination with blue standing for positive and orange for negative.
**a** Resonant interaction term appearing as $^2G_1^{23}$. **b** Resonant interaction term
appearing as $^2G_1^{43}$. Finally, **c** is a nonresonant term and **d** $^2G_1^{24}$ is a forbidden term (it
does not appear). These new interaction terms can be predicted from the combi-
natorics of the original network and coupling function.

The resonance condition Eq. (8) is satisfied for both. Using the repla-
cement rule Eq. (11), we find

$$u_k = z_k + \alpha \left( \frac{z_{k-1}z_k}{\bar{\gamma}_{k-1}} + \frac{z_k z_{k+1}}{\bar{\gamma}_{k+1}} + \frac{z_{k-1}^2 \bar{z}_k}{\gamma_{k-1} + \bar{\gamma}_k} + \frac{z_k^2 \bar{z}_{k+1}}{\gamma_k + \bar{\gamma}_{k+1}} \right) \qquad (14)$$

Each node equation contains 16 interaction terms as in Eq. (12). We
discuss some of these terms for the first node. $^2G_1^{23}$ appears as node 1 is
connected to node 2 and 2 to 3. This interaction is resonant, see Fig. 2a.
$^2G_1^{43}$ appears because node 1 is connected to 4 and node 4 to 3. This
term is also resonant, see Fig. 2b. $^1G_1^{24}$ is nonzero and nonresonant.
This term appear as 1 is directed connected to 2 and 4, see Fig. 2c.
Finally, the term $^2G_1^{24}$ is a forbidden, the term would appear from an
interaction of 1 to 2 and from 2 to 4, however, in the original network
the later interaction is absent, see Fig. 2d. Remarkably, not all inter-
actions are relevant when the goal is to describe slow oscillations in the
phases.

Indeed, once we analyse the phases in the new equations, the
coupling term coming from $^2G_1^{23}$ will lead to oscillations with fre-
quency close to $\omega_1 - \omega_2 + \omega_3$ while the term coming from $^2G_1^{43}$ leads
to a frequency close to $\omega_1 - \omega_4 + \omega_3$. This implies that both terms are
slowly varying. In contrast, the term coming from $^2G_1^{24}$ leads to
oscillations with frequency $\omega_1 - \omega_2 + \omega_4 \approx \omega_1$ and is fast oscillating in
comparison to the slow terms with small frequencies. In virtue of
the averaging theory, such fast oscillating terms can be neglected. In
fact, only resonant terms connected by local trees in the original
graph will survive such as the resonant ones involving $\omega_1 - \omega_2 + \omega_3$
and $\omega_1 - \omega_4 + \omega_3$. This yields

$$
\begin{aligned}
\dot{u}_1 &= f_1(u_1) - \alpha^2 \eta_{12} u_1^2 \bar{u}_2 u_3 - \alpha^2 \eta_{14} u_1^2 \bar{u}_4 u_3 \\
\dot{u}_2 &= f_2(u_2) - \alpha^2 \zeta_{231} u_2^2 \bar{u}_1 \bar{u}_3 \\
\dot{u}_3 &= f_3(u_3) - \alpha^2 \eta_{32} u_3^2 \bar{u}_2 u_1 - \alpha^2 \eta_{34} u_3^2 \bar{u}_4 u_1 \\
\dot{u}_4 &= f_4(u_4) - \alpha^2 \zeta_{431} u_4^2 \bar{u}_1 \bar{u}_3
\end{aligned}
\qquad (15)
$$

where $\eta_{pq} = \frac{1}{\gamma_p + \bar{\gamma}_q}$ and $\zeta_{pqr} = \frac{2}{\gamma_p + \bar{\gamma}_q} + \frac{2}{\gamma_p + \bar{\gamma}_r} + \frac{1}{\gamma_q} + \frac{1}{\gamma_r}$. Writing $u = re^{i\theta}$ we
obtain equations for the phases $\theta$. The averaging theorem gives

$$
\begin{aligned}
\dot{\theta}_1 &= \omega_1 - \alpha^2 r_0^3 \left[ \rho_{12}(\phi_1) + \rho_{14}(\phi_2) \right], \\
\dot{\theta}_2 &= \omega_2 - \alpha^2 r_0^3 \sigma_{231}(\phi_1) \\
\dot{\theta}_3 &= \omega_3 - \alpha^2 r_0^3 \left[ \rho_{32}(\phi_1) + \rho_{34}(\phi_2) \right], \\
\dot{\theta}_4 &= \omega_4 - \alpha^2 r_0^3 \sigma_{431}(\phi_2),
\end{aligned}
\qquad (16)
$$

where the phases $\phi_1$ and $\phi_2$ are given in Eq. (4). The functions $\rho$ and $\sigma$
are provided in the Supplementary Note 4. The emergent hypernet-
work explains the experimental fitting found in Eq. (5). These functions
represent hyperlinks as shown in Fig. 1f.

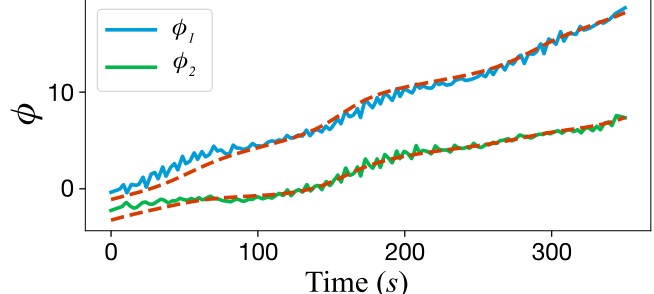

**Fig. 3 | Normal form theory explains the experimental results.** We show the time
series of the slow phase $\phi_1$ and $\phi_2$ from experimental data (solid) and the prediction
of the emergent hypernetwork (dashed) capturing higher-order interactions. The
vector field describing the phase interaction is obtained from first principles. The
coefficients of the vector field are obtained by least-square minimization.

The phase triplets $\phi_1$ and $\phi_2$ are revealed from phase reduction in
the normal form and they are not obvious from the original Eq. (6). We
confirm these predictions by direct simulations of Eq. (6) (Supple-
mentary Note 5). We present examples for a three-node path in Sup-
plementary Note 6 and a six-node network in Supplementary Note 7.

**Predicting the slow phase interactions in experiments**
In Supplementary Note 3, we show that the experimental recovery of a
hypernetwork is not an artifact. Rather, we prove that imposing spar-
sity unavoidably leads to the recovery of the normal form instead.
Indeed, as the recovery allows for a small least square deviation
between the data and the model, the recovery finds the hypernetwork
as a simpler description of the system. So, by measuring the original
variables and attempting a model recovery while imposing sparsity,
model recovery learns only the higher-order interactions. We now use
the emergent network prediction for the ring network with the cor-
responding resonance conditions as in the experiment to explain the
slow phase dynamics.

From the data we extract the slow phases $\phi_1$ and $\phi_2$ as shown in
Fig. 3 in solid lines. Using our theory, from Eq. (16), we obtain that

$$\dot{\phi}_i = \Omega_i + \sum_{j=1}^{2} a_{ij} \cos \phi_j + b_{ij} \sin \phi_j \qquad (17)$$

where $a$'s and $b$'s are given in terms of the functions $\sigma$ and $\rho$ in Eq.
(16) see Supplementary Note 5. We treat $a$'s and $b$'s as fitting para-
meters from the vector field in Eq. (17) obtained from first princi-
ples, since the corresponding coupling parameter and amplitudes
are unknown. The resulting solutions agree with the experimental
data as seen in Fig. 3. Our findings are not strictly limited to elec-
trochemical oscillators. As shown in Supplementary Note 9, we

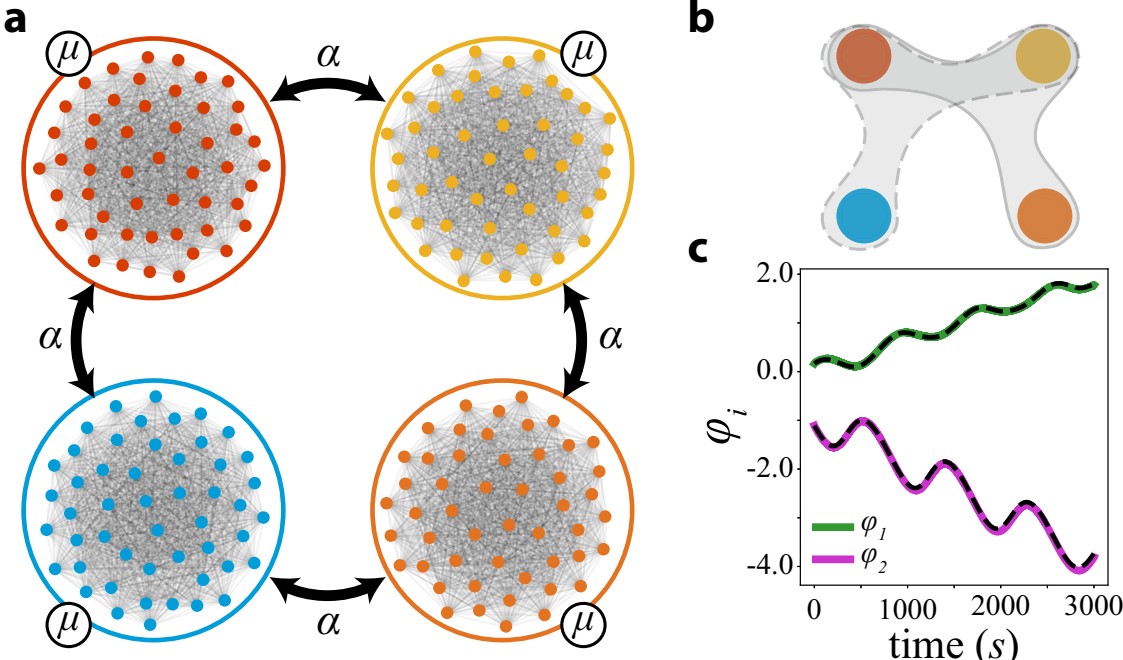

**Fig. 4 | Interacting subpopulations lead to higher order interaction of mean-fields. a** The original network of coupled subpopulations (with four distinct colours, namely, red, yellow, blue and orange). Oscillators are interacting by an internal coupling constant $\mu$ and inter-subpopulations coupling constant $\alpha$. **b** Higher order phase interaction of the mean-fields represented with the same colors as in **a** (red, yellow, blue and orange). Applying our approach we uncover

that the phase interaction between the mean-fields is described by a hypernetwork. **c** The mean-field slow phase variables $\varphi_1$ (green) and $\varphi_2$ (purple) were computed from the data collected from the simulations of mean fields on the associated network. The dashed curve is the simulation of the vector field of the slow phases $\varphi_{1,2}$ reconstructed from data using the Lasso method.

detected the same hypernetworks in nonlinearly coupled integrate-and-fire neuron models.

## Emergent hypernetworks among network modules coupled through mean-fields

The requirement of a nonlinear coupling, at first sight, seems to be a limitation for practical applications. However, here we analyze how hypernetworks emerge in modular networks with microscopic pairwise coupling through phase differences.

We consider four subpopulations of $N$ interacting Kuramoto oscillators[13]. Nodes in each subpopulation interact strongly among themselves with coupling strength $\mu$ and weakly between subgroups with coupling strength $\alpha$, see Fig. 4. As we will show at the macroscopic mean-field level, the interaction is nonlinear. According to our theory, although the mean-fields have a pairwise interaction, their model recovery will be in terms of hypernetworks. We first consider the microscopic description; each oscillator is described by

$$\dot{\psi}_{km} = \omega_{km} + \frac{\mu}{N}\sum_{n=1}^{N}\sin(\psi_{kn}-\psi_{km}) + \sum_{\ell=1}^{4}A_{kl}\left(\frac{\alpha}{N}\sum_{n=1}^{N}\sin(\psi_{ln}-\psi_{km})\right) \tag{18}$$

or in terms of mean-fields $\dot{\psi}_{km} = \omega_{km} + \mathrm{Im}\left(\mu z_k + \alpha\sum A_{kl}z_l\right)e^{-i\psi_{km}}$ where

$$z_k = \frac{1}{N}\sum_{m=1}^{N}e^{i\psi_{km}} \tag{19}$$

is the mean-field of the subpopulation $k$. The frequencies $\omega_{km}$ are distributed according to a Lorenzian $\rho(\omega, \Omega_k, \sigma_k)$ where $\Omega_k$ is the mean subpopulation frequency and $\sigma_k$ is the frequency dispersion.

Applying the Ott-Antonsen ansatz[15], we obtain the macroscopic equations describing the mean-fields in the limit $N \to \infty$ as

$$\dot{z}_k = f_k(z_k) + \sum_{l=1}^{4}A_{kl}h(z_k,z_l) \tag{20}$$

where $f_k$ is the Hopf normal form with constants $\gamma_k = (i\Omega_k + \mu - \sigma_k)$ and $\beta_k = -\mu$ and

$$h(z_k,z_l) = \alpha z_l + \alpha\bar{z}_l z_k^2, \tag{21}$$

thus, in the macroscopic description the coupling is nonlinear. We interpret $\alpha$ as a bifurcation parameter and deal with $\alpha z_l$ as a nonlinear term as in bifurcation theory. We consider the ensemble frequencies to satisfy the resonance conditions $\Omega_1 + \Omega_3 \approx 2\Omega_2$ and $\Omega_2 + \Omega_4 \approx 2\Omega_1$. At $\alpha = 0$ each subpopulation will have an order parameter behaving as $z_k(t) = r_k e^{i\theta_k(t)}$ where $r_k = \sqrt{\frac{\mu - \sigma_k}{\mu}}$ and $\dot{\theta}_k = \Omega_k$. To obtain the phase model, we bring the network to its normal form and apply the phase reduction. In Supplementary Note 10, we perform the calculations of such resonance conditions to obtain the new normal form equations. After discarding nonresonant terms the phase equations of the mean-fields read as

$$\begin{aligned} \dot{\theta}_{1,3} &= \Omega_{1,3} + F_{1,3}(\varphi_1) \\ \dot{\theta}_{2,4} &= \Omega_{2,4} + F_{2,4}(\varphi_2) \end{aligned} \tag{22}$$

where $F_i$ is a linear combination of sine and cosine.

Next, we fix the ensemble frequencies as $\Omega_1 = 2$, $\Omega_2 = 3$, $\Omega_3 = 4$ and $\Omega_4 = 1$ as well as the coupling strengths $\mu = 0.5$, $\sigma_k = 0.48$ yielding $r_k = 0.15$ and $\alpha = 0.1$ for all subpopulations. We numerically integrate the mean-field equations and obtain the complex fields $z_1(t)$, $z_2(t)$, $z_3(t)$ and $z_4(t)$ which enables us to extract the phase dynamics $\theta_1(t)$, $\theta_2(t)$, $\theta_3(t)$

and $\theta_4(t)$. Performing a Lasso regression we recover the vector fields of Eq. (22) confirming the theoretical prediction of higher order interactions, see Supplementary Note 10.

As before, we introduce the slow phases

$$\begin{aligned} \varphi_1 &= \theta_1 - 2\theta_2 + \theta_3, \\ \varphi_2 &= \theta_2 - 2\theta_1 + \theta_4. \end{aligned} \quad (23)$$

The theory predicts the higher order interaction between the slow phases as $\dot{\varphi}_k = \varepsilon_k + G_k(\varphi_1, \varphi_2)$, as shown in Supplementary Note 10. The fitting the predicted vector field of $\varphi$ to the data is excellent as can be observed in Fig. 4c.

For these four subpopulation on a ring, the condition on the frequencies is close to the subspace $V_{res} = \{\Omega_1 + \Omega_3 = 2\Omega_2, \Omega_2 + \Omega_4 = 2\Omega_1\}$, forming a co-dimension 2 resonance surface. That is, the emergence of hypernetworks is generic in a two parameter family of frequencies.

## Discussion

We have uncovered a mechanism by which nonlinear pairwise interactions with triplet resonance conditions result in nontrivial phase dynamics on a hypernetwork. Such interactions traditionally were attributed in brain dynamics to synaptic transmission between two neurons mediated by chemical messengers from a third neuron (heterosynaptic plasticity)[29]. Our findings provide an alternative mechanism. On one hand, this finding shows that phase dynamics can be mediated through 'virtual' interactions not physically present in the system. On the other hand, such a mechanism could be leveraged to design interactions between remote components not directly connected but instead having correlations in natural frequencies.

The experimental system with a generic network motif with a ring of four electrochemical oscillators presented here was an example, where a relatively simple nonlinear modulation of the coupling induced a hypernetwork driven phase dynamics. Networks with a ring topology are selected for the experiment since they are common for many network based complex systems, e.g., in lasers, biological systems, neuronal dynamics and many disciplines[30,31]. Such nonlinear modulation of the coupling can be quite general in gene expressions; for example, it was used to describe the coupling among circadian cells through Michaelis-Menten mechanism where coupling from one cell modulated the maximum gene expression rate in the other[32].

Strikingly, we showed that the coupling resulting in mean-field coupling among network modules has sufficient nonlinearity to facilitate hypernetwork interactions. In particular, event related modulation of spectral responses of magnetoencephalogram (MEG) recordings (i.e., modulation of frequency-specific oscillations in the motor network established by a handgrip task) have shown very strong evidence for nonlinear, between-frequency coupling of remote brain regions[33]. Our results strongly suggest that in these MEG recordings, given the appropriate resonances and nonlinearities, hypernetwork description could facilitate the long-range modulation of frequencies. In conclusion, the findings open new avenues for hypernetwork based description and engineering of complex systems with heterogeneous frequencies and nonlinear interactions.

## Methods

Our results give an algorithmic procedure for obtaining a hypernetwork that accurately describes the observed behavior of the original system. This emergent higher order system depends on details of the given network, the original coupling function and the resonance relations among the phases.

## Normal form calculations

In Supplementary Note 2, we consider ODEs of the general form

$$\dot{z}_k = \gamma_k z_k - \beta_k z_k |z_k|^2 + \alpha H_k(z_1, \ldots, z_n), \quad k \in \{1, \ldots, n\}, \quad (24)$$

with $z_k \in \mathbb{C}$ and $\alpha \in \mathbb{R}$. The numbers $\beta_k, \gamma_k \in \mathbb{C}$ are assumed non-zero, and we furthermore write $\gamma_k = \lambda + i\omega_k$. Here $\lambda \in \mathbb{R}$ is seen as the bifurcation parameter for a Hopf bifurcation, and we assume the interaction functions $H_k : \mathbb{C}^n \to \mathbb{C}$ to be smooth (i.e. $C^\infty$) for convenience. Moreover, we initially assume each $H_k$ satisfies $H_k(0) = 0$ and $DH_k(0) = 0$, though the condition on its derivative is later dropped.

Our main result shows that the ODE (24) can be put in a normal form that allows us to predict the phase dynamics of the oscillators. We do this by using two successive transformations:

$$w_k = z_k - \alpha P_k(z); \quad (25)$$

$$u_k = w_k - \alpha Q_k(w), \quad (26)$$

for some appropriately chosen polynomials $P_k$ and $Q_k$. The first of these coordinate transformations is used to remove the term $\alpha H_k(z)$ from the Eq. (24). This will generate additional terms in $\alpha^2$ that may be expressed in the coefficients of $H_k$ and $P_k$ following certain combinatorial rules. We manage this combinatorial behavior by introducing a special bracket $[\bullet||\bullet]$ on the space of polynomials. In addition to these new interaction terms, the transformation will also produce terms in $\alpha$ involving $P_k$ and $\beta_k z_k |z_k|^2$, which obscure an interpretation of the system as a (hyper) network. We therefore remove these additional terms using the second coordinate transformation. A crucial observation here is that the non-resonance conditions needed for the first transformation are sufficient to ensure the second. We are able to prove this using the precise bookkeeping enabled by the aforementioned bracket.

When dealing with the case where $DH_k(0) \neq 0$, we instead remove only the non-linear terms in $H_k$ using the transformations (25) and (26). This reveals higher order terms as before. Even though $DH_k(0)$ accounts only for nonresonant terms by assumption, this linear term will nevertheless cause an overall frequency shift that has to be accounted for. More precisely, if we denote by $\Omega$ the diagonal matrix with entries the frequencies $\omega_1, \ldots, \omega_n$, then the natural frequencies in the coupled case will be given by the imaginary part of the eigenvalues of $i\Omega + \alpha DH(0)$. Here we have set $H = (H_1, \ldots, H_n)$. These new frequencies can be approximated by standard eigenvalue perturbation techniques.

## Properties of the coupling functions $^1G_k^{\ell p}$ and $^2G_k^{\ell p}$

Applying the transformation of the theorem to Eq. (6) yields a new system of the form Eq. (12). In Supplementary Note 2, we show that

$$\begin{aligned} ^1G_k^{\ell p}(u_k, u_\ell, u_p) &= \frac{\partial \tilde{h}_{k\ell}(u_k, u_\ell)}{\partial u_k} h(u_k, u_p) + \frac{\partial \tilde{h}_{k\ell}(u_k, u_\ell)}{\partial \bar{u}_k} \overline{h(u_k, u_p)} \\ ^2G_k^{\ell p}(u_k, u_\ell, u_p) &= \frac{\partial \tilde{h}_{k\ell}(u_k, u_\ell)}{\partial u_\ell} h(u_\ell, u_p) + \frac{\partial \tilde{h}_{k\ell}(u_k, u_\ell)}{\partial \bar{u}_\ell} \overline{h(u_\ell, u_p)}. \end{aligned} \quad (27)$$

In Eq. (27) a term of degree $d$ in $h$ and a term of degree $\tilde{d}$ in $\tilde{h}_{k\ell}$ combine to form a term of degree $d + \tilde{d} - 1$ in $^1G_k^{\ell p}$. As both $h$ and $\tilde{h}_{k\ell}$ have terms of degree 2 and higher, we see that $^1G_k^{\ell p}$ only has terms of degree 3 and higher. The same holds true for $^2G_k^{\ell p}$, which means that a classical network description involving directed edges is no longer possible.

The third order terms are moreover easily found by replacing $h$ and $\tilde{h}_{k\ell}$ in Eq. (27) by their quadratic terms. Likewise, the fourth order terms are found by replacing $h$ by its quadratic terms and $\tilde{h}_{k\ell}$ by its cubic terms and vice versa in Eq. (27). We may also argue that these higher order terms in $^1G_k^{\ell p}$ and $^2G_k^{\ell p}$ are non-vanishing in general.

Indeed, the coefficients in front of these terms are rational functions of $\gamma_k$ and the coefficients of $h$. Such functions are either identical to the zero function (which Eq. (27) excludes) or non-vanishing on an open dense set.

New terms emerge that have an interpretation as higher-order interactions. The two double sums in Eq. (12) have a combinatorial interpretation. The first double sum counts all pairs of nodes $(\ell, p)$ that both influenced node $k$ in the original network. The second double sum counts all pairs $(\ell, p)$ where $\ell$ influenced $k$ and $p$ influenced $\ell$ and $p$ need not influence $k$ directly in the old network, so that new node-dependency is formed.

### An explicit algorithm for predicting the emergent hypernetwork

We present an algorithm for obtaining an emergent hypernetwork from a given network system. Its input consists of the adjacency matrix $A$, the function $h$ and the phases $\omega_1$ through $\omega_n$, and we assume the nonresonance conditions of the theorem to hold. The algorithm is as follows:

**Algorithm 1**. Emergent Hypernetworks

 **Input**: Adjacency matrix $A$, coupling function $h$, frequencies and amplitudes $\gamma_i$'s

 **Output**: Hypernetwork and Coupling functions

1: **for each** $k \in \mathscr{S}$ **do**
2: **for each** $\ell \in \mathscr{S}$ **do**
3: **if** $A_{k\ell} \neq 0$ **then**
4: form the polynomials $\tilde{h}_{k\ell}(u_k, u_\ell)$ by the replacement rule
$$z^{d_1} \bar{z}^{d_2} w^{d_3} \bar{w}^{d_4} \mapsto \frac{z^{d_1} \bar{z}^{d_2} w^{d_3} \bar{w}^{d_4}}{(d_1 - 1)\gamma_k + d_2\gamma_k + d_3\gamma_\ell + d_4\bar{\gamma}_\ell}$$
5: **for each** $p \in \mathscr{S}$ **do**
6: **if** $A_{k\ell}A_{kp} \neq 0$ **then**
7: Compute ${}^1G_k^{\ell p}$
8: **if** $A_{k\ell}A_{\ell p} \neq 0$ **then**
9: Compute ${}^2G_k^{\ell p}$
10: **procedure** Resonant terms in the coupling functions $G$
11: **for each** $u_k^{d_1} \bar{u}_k^{d_2} u_\ell^{d_3} \bar{u}_\ell^{d_4} u_p^{d_5} \bar{u}_p^{d_6}$ monomial of ${}^1G_k^{\ell p}$ and ${}^2G_k^{\ell p}$ **do**
12: **if** $(d_1 - d_2 - 1)\omega_k + (d_3 - d_4)\omega_\ell + (d_5 - d_6)\omega_p \neq 0$ **then**
13: discard term
14: **procedure** Remaining monomials are the couplings of node $k$

## Data availability

We provide the experimental time-series and the extracted phases of the oscillations (Fig. 1) at ref. 34. Source data are provided with this paper.

## Code availability

The source code for reconstructing the functions representing hypernetwork dynamics from oscillatory networks dynamics is available ref. 35.

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

## Acknowledgements

We thank Sajjad Bakrani, Zachary G. Nicolaou, Marcel Novaes, Edmilson Roque, Robert Ronge and Jeroen Lamb for enlightening discussions. T.P. was supported in part by FAPESP Cemeai Grant No. 2013/07375-0 and is a Newton Advanced Fellow of the Royal Society NAF\R1\180236. T.P. and E.N. were partially supported by Serrapilheira Institute (Grant No. Serra-1709-16124). D.E. was supported by TUBITAK Grant No. 118C236 and the BAGEP Award of the Science Academy. JLO-E acknowledges financial support from CONACYT. I.Z.K. acknowledges support from National Science Foundation (grant CHE-1900011).

## Author contributions

E.N. and T.P. designed the overall study and formulated the theory. J.L.O.-E. and I.Z.K. designed and performed the experiments. D.E. implemented the numerical simulations and analyses. All authors contributed to the writing of the manuscript. All authors reviewed and approved the final manuscript.

## Competing interests

The authors declare no competing interests.
