## [Peer Review File · Nature Communications]

REVIEWER COMMENTS

Reviewer #1 (Remarks to the Author):

In this work the authors propose a mechanism whereby hypernetworks with three-body coupling terms appear "naturally". Explanation is provided in terms of resonances that occur as a consequence of the coupling terms and the underlying network.

I found the paper a bit difficult to read since the results are presented mainly in the supplementary material. (I would encourage the authors to move some of it from there to the main paper; be kind to the readers.) It would also benefit from some editing to improve the language - some of the usage is very nonstandard and not always easy to decipher.

Here are some specific queries:

1. How were the systems actually coupled? If I understand correctly, each of the cells is isolated, so the coupling is done by extracting an electrochemical signal from each cell and feeding it into the cell it is coupled to. Is that correct? It would be good to know. Also, how is time-delay incorporated in this scheme?
2. How is Eq. 7 obtained?
3. If "Most strikingly, in Supplementary Material VIII, we show that the experimental recovery of a hypernetwork is not an artifact", it deserves some more prominence in the paper.
4. It is not a priori clear why the synchronization that might occur from triplet interactions is different from the results of this paper.

I missed, somehow, the wider applicability of this methodology and the significance of these results. I know that the authors have bigger applications in mind, but this has not come across at all. I know some reference is made to circadian coupling, etc. but this does not seem convincing.

Reviewer #2 (Remarks to the Author):

In their manuscript "Emergent hypernetworks in nonlinearly coupled networks" the authors present both experimental and theoretical results that uncover higher-order interactions in systems of coupled oscillator networks. (Experiments are composed of electrochemical oscillators, while theoretical results consider normal form-type dynamics for Hopf bifurcations.) The results on both ends are very strong. Beyond simply serving as a sort of "proof" that higher-order interactions exist and affect the dynamics, the current work lends important insight and makes strong contributions in terms of detailing what these interactions look like, both in terms of where they appear structurally and how they behave dynamically. After carefully picking through the theoretical parts I see no errors. The potential impact of this work is considerable, given the current interest in higher-order interactions and dynamics, and this work has the potential to significantly reshape this thrust of work moving forward. Overall, I recommend publication.

Reviewer #3 (Remarks to the Author):

the paper studies a rather special electrochemical experiment where higher order interactions between the oscillators emerge.

They then develop a formal model based on normal form computations which lead to apparent higher order interactions even when those are not directly physically present. Such computations are not really novel, the experiment is rather special, and the scope of the findings does not seem as wide as the authors claim.

In summary, it is an interesting and competent contribution, but certainly not at the level of Nature Communications.

REVIEWER COMMENTS

Reply to Reviewer #1:

In this work the authors propose a mechanism whereby hypernetworks with three-body coupling terms appear "naturally". Explanation is provided in terms of resonances that occur as a consequence of the coupling terms and the underlying network.

We thank the referee for the careful and insightful review of our manuscript. We address all concerns of the referee below.

I found the paper a bit difficult to read since the results are presented mainly in the supplementary material. (I would encourage the authors to move some of it from there to the main paper; be kind to the readers.) It would also benefit from some editing to improve the language - some of the usage is very nonstandard and not always easy to decipher.

Following the suggestion, we brought some material to the main text, namely:

- i) the setting of the main theorem and its discussion is presented in Methods, describing the techniques of the proof on normal form reduction in greater detail.
- ii) The explanation of Eq. (7) is provided in lines 109 and 125
- iii) We added a discussion that resonance conditions alone do not predict the resulting emergent network in lines 87-91. We also included a discussion on the new coupling functions and averaging in lines 159-166.

Here are some specific queries:

1. How were the systems actually coupled? If I understand correctly, each of the cells is isolated, so the coupling is done by extracting an electrochemical signal from each cell and feeding it into the cell it is coupled to. Is that correct? It would be good to know. Also, how is time-delay incorporated in this scheme?

Yes, it is correct that the electrodes are only negligibly coupled inherently, i.e., through the potential drops in the electrolyte (=isolated), and the coupling is done by feeding the signals into the cell with a delayed feedback scheme. Now we state this explicitly in figure caption 1 and also explain that the time delay is introduced by storing the past recordings in the memory of the computer. The detailed scheme of the coupling is described around Eqs 1 and 2.

2. How is Eq. 7 obtained?

We added an explanation in lines 109 and 125.

3. If "Most strikingly, in Supplementary Material VIII, we show that the experimental recovery of a hypernetwork is not an artifact", it deserves some more prominence in the paper.

We expanded this discussion in lines 178-181.

4. It is not a priori clear why the synchronization that might occur from triplet interactions is different from the results of this paper.

Before one can perform extensive (hyper)synchronization studies, the dynamical features have to be explored and the mathematical machinery has to be developed that allows such studies. Synchronization from triplet interactions would require the development of phase models through hyperLaplacian interactions, and a method to analyze the synchronization. This is certainly interesting, and our group members are working on this problem in a separate research direction. In this current manuscript, our focus was how the dynamics can be described, i.e., how the frequency modulation equations will emerge, which triplet interactions appear, and which are forbidden. Consequently, researchers now will be able to write realistic hypernetwork equations, and thus synchrony patterns can be analyzed. In other words, our presented work sets the stage for future studies of realistic hypersynchronization patterns.

I missed, somehow, the wider applicability of this methodology and the significance of these results. I know that the authors have bigger applications in mind, but this has not come across at all. I know some reference is made to circadian coupling, etc. but this does not seem convincing.

In the revised manuscript, we made two important breakthroughs, which demonstrate the wide applicability of the results. We found that hypernetwork emerge when

- A. Modules couple through mean-fields using Kuramoto phase oscillators
- B. Integrate-and-fire model neurons coupled on a ring.

These findings allowed us to give an even broader scope of possible applications. Most importantly, it is now clear that modular neural circuits are expected to generate modulation of frequencies due to emergent hypernetworks. In the Conclusions, we cite research on MEG recordings, where coupling across different frequency regions is nonlinear (J. Neurosci., 10.1523/jneurosci.1194-09.2010). This, along with our findings, can result in an exciting application, where neuronal firing frequencies can be modulated (and thus neurotransmitters expressed rhythmically) as a result of nonlinearity (either direct or indirect due to mean field interactions) and resonances. This would be an alternative mechanism to current models of chemical regulations in biochemical networks, which occur through heterosynaptic plasticity. This new regulation scheme is now described in the Conclusions section.

Reply to Reviewer #2:

In their manuscript "Emergent hypernetworks in nonlinearly coupled networks" the authors present both experimental and theoretical results that uncover higher-order interactions in systems of coupled oscillator networks. (Experiments are composed of electrochemical oscillators, while theoretical results consider normal form-type dynamics for Hopf bifurcations.) The results on both ends are very strong. Beyond simply serving as a sort of "proof" that higher-order interactions exist and affect the dynamics, the current work lends important insight and makes strong contributions in terms of detailing what these interactions look like, both in terms of where they appear structurally and how they behave dynamically. After carefully picking through the theoretical parts I see no errors. The potential impact of this work is considerable, given the current interest in higher-order interactions and dynamics, and this work has the potential to significantly reshape this thrust of work moving forward. Overall, I recommend publication.

We thank the reviewers for the time evaluation of the paper and for the recommendation. We added further examples to illustrate the applicability of the approach.

Reply to Reviewer #3:

the paper studies a rather special electrochemical experiment where higher order interactions between the oscillators emerge. They then develop a formal model based on normal form computations which lead to apparent higher order interactions even when those are not directly physically present.

We thank the reviewer for the time spent evaluating our manuscript. We stress that the higher-order interactions are not “apparent”, Our theory comes from bifurcations. This means that for small parameters such as coupling strength and radius of the orbits, the theory states as long as the sparsity is imposed the original equations are lost and only the higher order appears. In fact, we demonstrate mathematically and experimentally that hyper-networks appear unavoidably in network recovery.

We call attention to two key contributions we provide:

1. It is impossible to reconstruct the pairwise interactions if sparsity is imposed.
2. The emergent higher-order connections cannot be predicted based on resonance conditions alone. Instead, there is an intricate interplay between resonance and network topology, which depends on the details of the coupling function. Our algorithm provides an efficient way to obtain them. In particular, in the experiment, the pairwise resonance does not play a role.

Such computations are not really novel, the experiment is rather special, and the scope of the findings does not seem as wide as the authors claim.

We call the attention to the following 3 points

1. Novelty of computations: Our theory generalizes the classical normal form to the network setting. This is highly novel. Trying to eliminate coupling terms will generate new terms via the isolated dynamics. We are able to develop a technique involving multiple controlled transformations and a novel bracket on polynomial space that keeps track of new terms appearing and eliminates them from the isolated dynamics.
2. Experiment specialty: The experiments utilize four chemical oscillators under a delayed feedback; similar experiments are often used to demonstrate highly novel synchronization phenomena, for example, BZ beads with light feedback (Nat Phys 10.1038/s41567-017-0005-8), or liquid-crystal with a spatial light modulator (Nat Phys, 10.1038/nphys2372). The experiments were rather robust, and did not require optimization of parameters outside the resonance frequency conditions in accordance with the theory.

3. Scope of findings: In the revised manuscript, we made two important additional breakthroughs: We found that emergent hypernetworks occur in

A. Modules coupled through mean-fields using Kuramoto phase oscillators

B. Four integrate-and-fire model neurons coupled on a ring.

These findings allowed us to give a very broad scope of possible applications. Most importantly, we found literature evidence that distant brain regions are nonlinearly coupled across frequency regions (J. Neurosci., 10.1523/jneurosci.1194-09.2010). This evidence, combined with our results, implies that neural circuits could be expected to generate modulation of frequencies due to emergent hypernetworks. Consequently, one can predict that in such examples neuronal firing frequencies can be modulated (and thus neurotransmitter expressed rhythmically) as a result of nonlinearity (either direct or indirect due to mean field interactions) and resonances. This would be an alternative mechanism to current models of chemical regulations in biochemical networks, which occur through heterosynaptic plasticity. This new broader scheme is now described in the Conclusions section.

With these major improvements, we are convinced that the manuscript is at the level of Nature Communications.